# Pediatric Interventional Neuroradiology: Opportunities and Challenges

**DOI:** 10.3390/children10040715

**Published:** 2023-04-12

**Authors:** Cene Jerele, Dimitrij Lovrič, Dimitrij Kuhelj

**Affiliations:** 1Clinical Institute of Radiology, University Medical Centre Ljubljana, Zaloška Cesta 7, 1000 Ljubljana, Slovenia; 2Faculty of Medicine, University of Ljubljana, Vrazov Trg 2, 1104 Ljubljana, Slovenia

**Keywords:** pediatric interventional radiology, endovascular procedures, neuroradiological interventions, pediatric-specific interventional techniques, radiation protection

## Abstract

Pediatric interventional neuroradiology (PINR) is a relatively new field of diagnostic and therapeutic care in the pediatric population that has seen considerable advances in recent decades. However, it is still lagging behind adult interventional neuroradiology due to a variety of reasons, including the lack of evidence validating pediatric-specific procedures, the relative absence of pediatric-specific equipment, and the challenges in establishing and maintaining PINR competencies in a relatively small number of cases. Despite these challenges, the number and variety of PINR procedures are expanding for a variety of indications, including unique pediatric conditions, and are associated with reduced morbidity and psychological stigma. Continued technological advances, such as improved catheter and microwire designs and novel embolic agents, are also contributing to the growth of the field. This review aims to increase awareness of PINR and provide an overview of the current evidence base for minimally invasive neurological interventions in children. Important considerations, such as sedation, contrast agent use, and radiation protection, will also be discussed, taking into account the distinct characteristics of the pediatric population. The review highlights the usefulness and benefits of PINR and emphasizes the need for ongoing research and development to further advance this field.

## 1. Introduction

Pediatric interventional neuroradiology (PINR) is a relatively new field that has been expanding with new technological advancements in the field of interventional radiology. It is addressing diagnostics and lately also therapeutic approaches in the pediatric population [1]. We have been witnessing considerable advancement in recent decades; however, it is still lagging behind adult interventional neuroradiology for a variety of reasons. These include the lack of evidence validating pediatric-specific procedures, the relative absence of designated pediatric equipment, as well as continuity in maintaining standards of PINR in a relatively small number of cases [2]. PINR practice is often limited to a few specialized centers with a small number of trained professionals performing the procedures, limiting the availability of this type of patient care.

Nonetheless, the use of PINR procedures is increasing in many different pathological conditions. The indications include many of those found in the adult population as well as uniquely pediatric conditions that occur either in isolation or as part of different syndromes. Earlier treatment of certain types of congenital pathologies, such as low- and high-flow vascular malformations, can stop their propagation, reduce symptoms, and diminish the psychological burden related to them. PINR procedures are, by definition, minimally invasive, reducing the burden on the patient and minimizing intra- and post-procedural morbidity. Continuous technological development over time, including catheter and microwire designs, the evolution of novel embolic agents, and enhanced angiographic imaging solutions, leads to the conversion of classical surgical therapies into less invasive PINR solutions.

This review will cover the current evidence base for minimally invasive neurological interventions in children, reflecting the rapid growth and increasing demand for PINR procedures. Additionally, important considerations such as sedation, contrast agent use, and radiation protection will be discussed, taking into account the distinct characteristics of the pediatric population. The aim is to increase awareness of the field of PINR and familiarize readers with its usefulness and benefits.

## 2. Neurointerventional Procedures in the Pediatric Population

This chapter will include the basics of the clinical presentation, diagnosis, and treatment of several pediatric conditions that are most commonly referred to the interventional neuroradiologist for treatment.

### 2.1. Vascular Malformations

Vascular malformations are the most common diagnosis referred to the PINR department. According to the ISSVA (International Society for the Study of Vascular Anomalies) classification, they are generally divided into simple and combined malformations [3]. The former is more common and includes arteriovenous malformations (AVMs), venous, lymphatic, and capillary malformations, as well as arteriovenous fistulas. While vascular malformations such as AVMs can present in both children and adults, more specific pediatric vascular malformations include malformations of Galen’s vein, malformations of the dural sinus, and juvenile spinal vascular malformations.

#### 2.1.1. Arteriovenous Malformations

An AVM is presented as a complex network of vessels (nidus) with a direct connection between feeding arteries and draining veins without intervening capillaries. Symptoms of AVMs depend on the location and size of the AVM and can clinically present as tissue overgrowth, hyperemia, pain, pulsatility, tissue loss, bleeding, and rarely high-output heart failure [4]. For the diagnosis of AVMs in children, noninvasive imaging modalities such as magnetic resonance imaging (MRI) should be prioritized [5]. However, difficult computerized tomography angiography (CTA) and/or digital subtraction angiography (DSA) still present viable options (Figure 1).

Interventional treatment of AVMs is based on embolization [6,7]. It can be performed using either an arterial, venous, or combined approach. In the arterial approach, occlusion of the AVM’s arterial blood supply should target the nidus and spare as much as possible the normal branches to avoid unnecessary damage to the surrounding tissue that is often irrigated by the same branches. Consequently, the procedures are very challenging, requiring extensive knowledge and understanding of vascular anatomy. Angiography must be performed diligently to determine the feeding arteries, which are then super-selectively catheterized, and embolized. Embolic agents come in a variety of forms, including liquid (e.g., ethanol), semiliquid (e.g., NBCA glue and Onyx), and solid (e.g., particles, gelfoam, and coils) [8,9]. Transvenous occlusion of the nidus can represent a safer and viable option in lesions with accessible dominant draining veins in comparison to the transarterial route. The approach to each lesion should be individualized according to its location and structure. Ideally, the procedure is performed by combined embolization of the feeding arteries, nidus, and draining veins. The risk of complications, including tissue ischemia and nontarget embolization, may be reduced by staged sessional treatment [10].

#### 2.1.2. Vein of Galen Malformation

Vein of Galen malformations (VOGMs) are rare intracranial anomalies, accounting for less than 1–2% of all intracranial vascular malformations [11,12]. However, the mortality rate in the absence of treatment is nearly 100% [13,14,15]. With ultrasound becoming widely available, the diagnosis is increasingly made antenatally with findings of hydrocephalus, intracranial hemorrhage, or cardiomegaly [16,17]. The decision for treatment is made using a grading system of severity based on cardiac, cerebral, respiratory, hepatic, and renal function [18].

Transarterial embolization has been proven to be a very successful procedure, showing an overall survival rate of 76.9% and two-thirds of patients demonstrating no significant neurodevelopmental delay [19]. It has quickly become the treatment of choice, with surgical treatment typically having high morbidity and mortality [20]. Embolization is usually done with a transarterial approach using a similar technique to that in AVM treatment (Figure 2). Due to the very young age of the VOGM patients, vascular access can present a challenge. To avoid catheterizing and potentially traumatizing the tiny femoral arteries, the trans-umbilical artery approach can be used, offering straightforward access to the aorta in an atraumatic fashion [21]. Various embolic agents have proven successful in the treatment of VOGM, including coils and glue [18].

#### 2.1.3. Venous Malformations

Venous malformations (VMs) are the most common vascular anomalies that occur in the head and neck area in approximately 40% of cases [22]. Clinically, superficial VMs appear as bluish, compressible, and non-pulsatile lesions of the involved skin, which may grow over time [23]. While the initial diagnosis of VMs is usually clinical, the majority of lesions’ size, extent, and type are depicted by a Doppler ultrasound and an MRI examination [24,25]. Treatment is considered when VMs are causing pain or significant cosmetic problems and should be initiated early in life since fewer procedures and smaller volumes of sclerosants are required [4].

Fluoroscopic and/or ultrasound-guided percutaneous sclerotherapy is considered the first-line treatment for VM [26,27]. Initially, contrast agents are commonly used to delineate the full extent and volume of the lesion. Subsequently, one of the several different sclerosing agents (ethanol, sodium tetradecyl sulfate, etc.) can be utilized for the performance of sclerotherapy [28,29]. Sclerotherapy is typically performed in sterile conditions, and general anesthesia might be required in younger populations. Depending on the extent of the lesion, some patients may require multiple sessions over time.

### 2.2. Ischemic Disease

Pediatric stroke is a rare entity [30]. Due to its different and often non-specific manifestation compared to adults, it is often diagnosed with significant delay [31,32,33]. The most common risk factors for the occurrence of stroke in children are vasculopathy, infections, congenital and acquired cardiac disease, and coagulopathies [34]. As in adults, CT imaging is the modality of choice for acute presentations of focal neurologic deficits in children, with CT angiography being performed in a suspected large vessel occlusion [35,36].

In acute ischemic stroke, mechanical thrombectomy can be a valuable method to revascularize large-vessel cerebral arterial occlusion (Figure 3) [37]. Due to the extremely low caseload, mechanical thrombectomy in childhood stroke is not well established. However, current literature shows a high success rate of revascularization in children as well as short- and long-term treatment outcomes [38,39,40,41]. A variety of treatments were applied, including stent retrievers, the Penumbra system, and other mechanical devices [40]. Successful treatment is therefore based on symptom recognition and a timely diagnosis of pediatric stroke. The decision for endovascular treatment is best undertaken on a case-by-case basis at high-volume pediatric cerebrovascular centers [42].

### 2.3. Aneurysms

The discovery of intracranial aneurysms in children is rare and mostly incidental [43]. The pathogenesis is commonly related to underlying genetic abnormalities and developmental vascular tissue defects since the majority of the risk factors associated with intracranial aneurysms in adults are non-existent [44,45]. Cerebral aneurysms in children are morphologically different from their saccular counterparts in adults, presenting mostly in fusiform shape, dissecting, giant in size, and de novo aneurysms [46]. Clinical presentation of childhood pediatric aneurysms is variable, including subarachnoid hemorrhage, headache, direct compressive effects, focal neurologic deficits, or seizures. While CT and MR imaging are usually the initial choices in aneurysm diagnosis, DSA remains the gold standard for visualization and potential treatment planning (Figure 4) [47].

In recent years, endovascular therapy has become the treatment of choice for pediatric aneurysms. The femoral approach is mostly used, followed by catheterization and cerebral angiography, which are similar to those in adults. The most commonly used techniques are coiling with or without additional techniques such as balloon remodeling or stent implantation. The use of newer tools such as flow-diverting stents and intravascular flow-disrupting devices is not well-established in children but may present a valid treatment option [48]. In children, special considerations include careful blood loss monitoring and fluid replacements.

### 2.4. Tumors

Locoregional treatment of CNS tumors in children has been mostly limited to hypervascular intracranial and extracranial head, neck, and spinal tumors [49]. Tumor hypervascularity can be depicted on diagnostic imaging and differs between the tumor types. PINR offers embolization, often as an adjunct to preoperative management in order to decrease blood loss and facilitate tumor resection. In selected cases, it can also provide tumor-related pain reductions, decreased tumor progression, or reduced tumor-related bleeding [42]. It is often performed by transarterial embolization with liquid agents or particles or occasionally with coil implantation into feeding vessels. Hypervascular pediatric brain tumors such as choroid plexus papillomas, meningiomas, astrocytomas, hemangioblastomas, yolk sac tumors, and skull base tumors can be embolized preoperatively to reduce blood loss (Figure 5) [50,51]. However, the data is controversial, as a recent systematic review showed that endovascular embolization was found to be ineffective in reducing intraoperative blood loss, reducing surgical risk, or decreasing complications. Furthermore, in this series, intracranial endovascular embolization itself introduced significant risk, and complications were found to occur in more than 10% of patients [52]. Apart from the careful considerations of the risk versus benefits for each case, other relative contraindications for the intravascular tumor treatment include allergy to iodinated contrast agents, severe renal dysfunction, occlusion of feeding vessels, restricted access (e.g., femoral/iliac artery occlusion), and challenging regional vascular anatomy that may prevent the selective catheterization.

### 2.5. Intraarterial Chemotherapy for Retinoblastoma

Retinoblastoma (RB) is the most common intraocular malignancy in children [53,54]. If left untreated, this aggressive cancer is fatal; however, early diagnosis and advances in its treatment may result in a highly curable disease. Traditionally, the disease was managed with combined systemic chemotherapy, irradiation, focal treatments (cryotherapy, thermotherapy, etc.), or globe enucleation. Precise delivery of chemotherapeutic agents into the ophthalmic artery, available in the last decade, became the preferred treatment [55,56]. This targeted strategy has been shown to improve ocular survival and visual acuity while avoiding the severe systemic complications associated with systemic chemotherapy or the use of irradiation therapy [57]. The most commonly used chemotherapy agent in RB treatment is currently Melphalan combined with local treatment to achieve a synergistic effect [58]. Despite the risks of ocular vasculopathy and radiation toxicity, the procedure is promptly used for early and advanced cases [53].

### 2.6. Petrosal Sinus Sampling

Cushing’s syndrome diagnostic workup is often determined by simultaneous bilateral inferior petrosal sinus sampling, being the most accurate procedure in the differential diagnosis of hypercortisolism of pituitary or ectopic origin, compared to clinical, imaging, or biochemical analyses [59]. The procedure is performed through a bilateral femoral vein approach. Catheters are placed in the inferior petrosal sinuses bilaterally, and venous blood samples are collected before and after the administration of the corticotrophin-releasing hormone. When performed by an experienced interventional radiologist, a high success rate is usually achieved, as reported in a study with a success rate in more than 90% of cases [59]. However, due to its cost and invasiveness, the procedure is only used in patients in whom the cause cannot be localized based on non-invasive imaging findings in the presence of laboratory and clinical evidence of Cushing’s syndrome [60].

## 3. Special Considerations for Pediatric Patients

### 3.1. Sedation/Anesthesia

Adults mostly tolerate interventional procedures, since they are usually able to understand their goal. In the pediatric population, performing PINR procedures presents unique challenges related to sedation and anxiolysis. The sedation levels in this population might be higher to reduce motion, increase pain tolerance, and allow stable positioning during the intervention [61,62]. Deep sedation or general anesthesia also provides children with post-procedure amnesia. Drugs should be dosed in pediatrics on a per-kilo basis, with recognition of the differences in pharmacokinetics and pharmacodynamics that pertain to children in contrast to adults [63]. Concerns have arisen regarding the utilization of anesthesia in infants and children, particularly after the U.S. Food and Drug Administration’s warning that the use of general anesthetics and sedation drugs repeatedly or during lengthy procedures may negatively affect brain development in children younger than 3 years [64]. Therefore, measures should be taken by PINR practitioners to avoid any unnecessary exposure to prolonged or repeated anesthesia as well as to apply all available strategies to reduce adverse effects when anesthesia cannot be avoided [65]. Knowledge of current recommendations is crucial for the PINR team, with the first and foremost concern being the safety of the child.

### 3.2. Radiation

Interventional fluoroscopy procedures, and especially lengthier and more complex neurointerventional procedures, require extensive irradiation due to fluoroscopic guidance and large numbers of radiographic images. Radiation exposure poses a valid concern, since not only the possibility of stochastic effects but also deterministic levels might be reached [66,67,68]. Children require special attention because they are more sensitive to radiation and have longer lifespans to express changes [69,70,71,72]. A study by Thierry-Chef et al. found that the lifetime risk of brain tumor diagnosis in pediatric patients was estimated to be increased over the normal background rates by 3–40% depending on the dose received, age at exposure, and gender [73].

The risk of radiation exposure should always be balanced by the benefits of a given radiological procedure and its potential damage, and every measure must be taken to reduce the radiation dose as much as possible. The principles of “As Low As Reasonably Achievable” (ALARA), “Image Gently”, “Step Lightly”, and “Pause and Pulse” should be followed [74]. Additional precautions should be implemented, for example minimizing fluoroscopy time, using collimation, lowering the frame rate, holding the last image, and using digital zoom whenever possible [63,75]. Efforts should be made to avoid any unnecessary radiation of the pediatric eye or gonads. Since PINR is a team effort, all the players should be aware of their role in permanent monitoring of the fluoroscopy time, radiation dose, and image optimization and alerting the interventional radiologist, if needed, to achieve the lowest possible exposure.

### 3.3. Contrast Agent

Contrast agents (CA) must be used sparingly in children and their use must be closely monitored. Diluted low-osmolar non-ionic contrast should be used to minimize renal toxicity. The doses should be limited to 4–5 mL/kg for neonates and 6–8 mL/kg for older children [76]. In addition, dilution of CA with saline in a 50:50 or lower ratio should be used to limit the volume of CA used. The use of smaller syringes (3 or 5 mL) optimizes the amount of CA applied as well as the amount of fluid injected in small children [63]. In neonates, aspirating the contrast from the dead space of the catheter reduces the contrast load. Fluid overload, blood loss, and hypothermia from excessive saline solution flush and blood aspiration may, especially in neonates, present a serious threat to vital functions. Continuous monitoring of the CA usage should be implemented during the procedure, including fewer automatic injectors and more hand injections. Similar to radiation protection, CA volume should be monitored by the team and the operator should be alerted to cumulative volume excess used during the procedure.

### 3.4. Equipment

The expansion of PINR only partially transfers into the innovation of pediatric-specific devices. Shorter and smaller diameter guidewires and vascular access catheters play important roles in designated populations and also in complication reduction. Still, the majority of the material used in pediatric patients is adapted from the adult world, where the profiles are getting smaller and the flexibility and deliverability of catheters are increasing. There is a constant improvement in the stent, vascular occlusion, and angioplasty balloons technique, increasing the possibility of their usage in the pediatric population. In the ideal setting, the interventional radiologist has time to plan and order devices that are tailored to pediatric patients. When facing limited time notice and/or the lack of pediatric-sized equipment, many interventional radiologists will use or adapt adult-sized equipment that is readily available. Catheters may be shortened, ultrasound guidance for vascular access should be implemented, and the smallest needles, compatible with adequate guidewires, sheaths, and catheters, should be used whenever possible [63]. Due to the limited number of cases performed in PINR, the extrapolation of procedure outcomes and adult-approved devices to the pediatric population is still to be evaluated, especially in the field of safety and efficacy [77]. Institutions performing PINR regularly should strongly consider expanding designated pediatric equipment to improve performance and reduce complications.

### 3.5. Vascular Access

Obtaining vascular access in the pediatric population presents a challenging part of PINR procedures. Several key differences exist between the adult and pediatric populations. Anatomically, pediatric vessels are located superficially, especially in neonates and small children, arteries are without intrinsic diseases and are lacking tortuosity, often present in the elderly. Catheterization of pediatric vessels can result in occlusion, especially due to larger catheter-to-vessel ratios in children weighing less than 15 kg. There is also a higher prevalence of vasospasm [78,79] and dissection [80] compared to adults.

As in adult interventional procedures, the common femoral artery is the standard access site in most arterial PINR procedures. Alternatively, axillar, brachial, and umbilical access might be required in certain patients. The umbilical artery represents direct access to the arterial system; however, it is usually patent for up to 5 days after birth and it can be used only in neonates or premature infants. Due to its size, it can accommodate large sheaths and catheters with sparing peripheral vasculature and potential assess-related complications [80,81,82]. Direct access to the carotid or vertebral artery was used in some centers; however, due to its complexity and potential complications, it should be reserved for very complex PINR procedures and high-volume centers [82].

Vascular access was traditionally achieved by palpation and may be challenging in the obese population, including children. With the introduction of ultrasonographic equipment (US) in angiographic rooms, it became a gold standard in accessing vasculature in adults in many centers. Direct vessel US visualization revolutionized vascular access and should be considered in all patients, not only those with difficult access, reduced pulses, and multiple previous catheterizations [76]. Standard linear transducers are most commonly used to visualize the desired puncture location. Infiltration of a local anesthetic agent before puncture could increase patients’ comfort, however, it should not obliterate the pulse. Failed attempts of vascular access often require manual compression for 1–5 min, depending on the size of the needle. Hemostasis should be performed gently to avoid prolonged vessel occlusion and to ensure distal perfusion of the extremity, minimizing the possibility of arterial thrombosis [76].

## 4. Conclusions

PINR is an expanding field based on technical advances in minimally invasive interventional radiology. With the emergence of the field being fairly recent, there are still many challenges that practicing interventional radiologists face. However, with increasing evidence validating pediatric-specific procedures and advances in endovascular technology, the PINR will likely become the standard in the multidisciplinary management of pediatric neurologic conditions. Conscious efforts must be made to raise awareness among clinicians as well as the general public about the tremendous advantages that PINR offers to their patients.

## Figures and Tables

**Figure 1 children-10-00715-f001:**
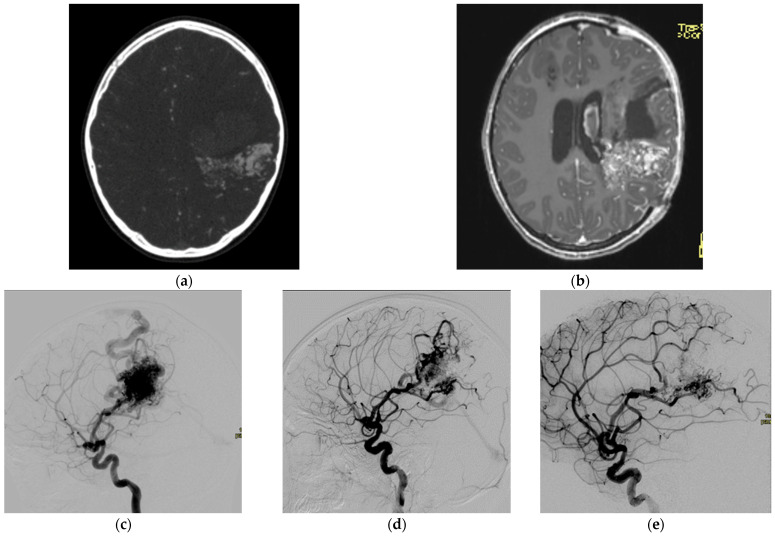
An 8-year-old female with an extensive AVM. AVM is shown on CTA (**a**), MRI (**b**), and DSA in sagittal projection (**c**). The AVM has been treated in staged sessional treatments using liquid embolic agent. The reduction in AVM size after the first treatment session (**d**) has been estimated at 60–80% compared to the starting size. There has been additional reduction in size after the second treatment session (**e**).

**Figure 2 children-10-00715-f002:**
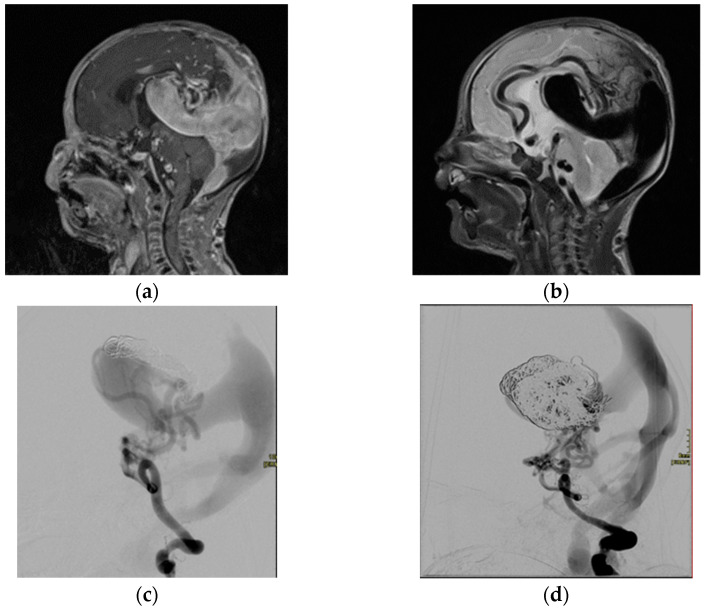
A 2-month-old infant with a choroidal type of Vein of Galen malformation as seen on contrast-enhanced T1 (**a**) and T2 MRI in the sagittal plane (**b**). The malformation has been progressively embolized in several staged sessional treatments (**c**,**d**). Coils have been used for embolization.

**Figure 3 children-10-00715-f003:**
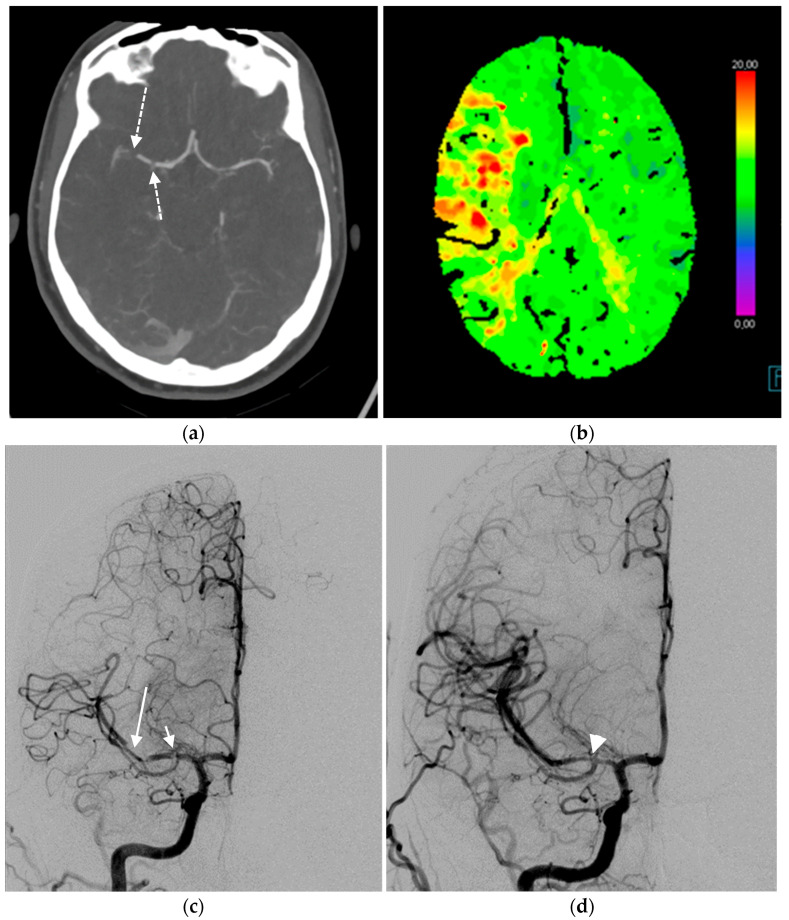
A 17-year-old female with acute ischemic stroke. The CTA showed an occluded right middle cerebral artery (dashed arrows, (**a**)) and increased time to peak perfusion maps in the corresponding area of the brain (**b**). The DSA showed two emboli in the course of the right middle cerebral artery (**c**). The shorter one (3 mm) was located in the middle of the M1 segment (short arrow), while the longer one (10 mm) was located in the distal part of the M1 segment, extending into the M2 segment (long arrow). Both emboli were removed through mechanical thrombectomy (**c**). Moderate vasospasm (arrowhead) was seen immediately after the removal (**d**) and subsequently subsided in a matter of minutes.

**Figure 4 children-10-00715-f004:**
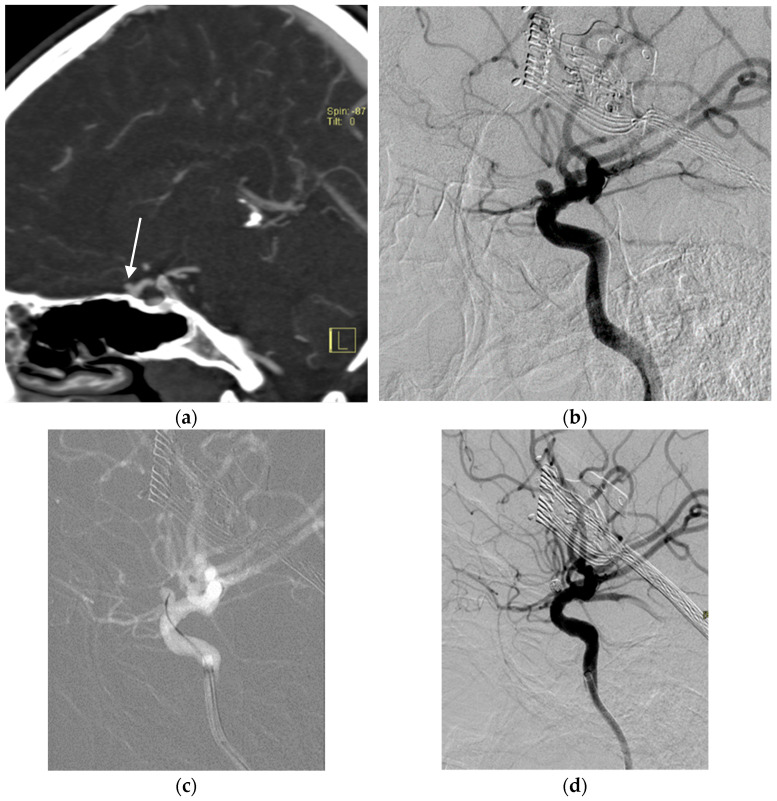
A case of intracranial aneurysm in a 16-year-old female. The CTA showed a 3 mm saccular aneurysm in the ophthalmic segment of the internal carotid artery (arrow, (**a**)). The DSA image confirms the aneurysm (**b**), which is subsequently treated with coils (**c**). The control DSA shows the excluded aneurysm (**d**).

**Figure 5 children-10-00715-f005:**
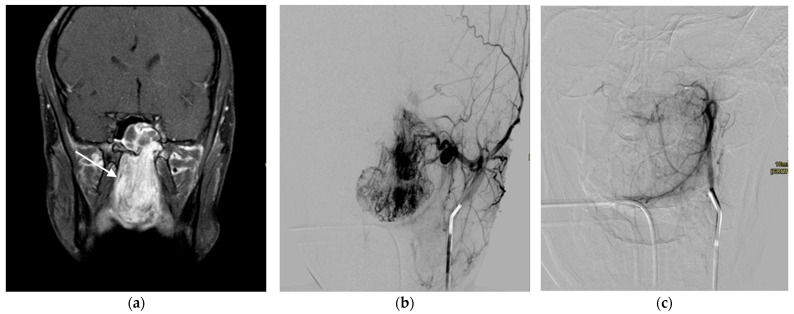
A 17-year-old male with juvenile nasopharyngeal angiofibroma (JNA). Preoperative embolization of JNA was performed. T1-weighed contrast-enhanced MRI sequence in the coronary plane shows JNA (arrow) that fills up most of the nasal cavity (**a**). Intraoperative DSA shows hypervascular tumor formation of the nasopharynx, nasal meatuses, and ethmoid cells (**b**). Superselective particle embolization was performed, and control DSA showed a good result (**c**).

## Data Availability

Data available on request due to restrictions e.g. privacy or ethical. The data presented in this study are available on request from the corresponding author. The data are not publicly available due to privacy protection.

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
