# Peer review of "Pediatric Interventional Neuroradiology: Opportunities and Challenges"

_children, 2023, doi:10.3390/children10040715_

Round 1

Reviewer 1 Report

I generally appreciated the manuscript. It is well organized and written in detail. 

I have some suggestions:

1. Diagnostic images about the presented pathologies should be given

a. A CTA and MRI image for AVM case 

b. If possible a MR venography image of Vein og Galen malformation

c. DWI -ADC- and CT images of the stroke case

d. an example of aneurism along with CTA images

e. An additional tumor case other than angiofibroma especially an intracranial tumor

f. a procedure example and diagnostic images for retinoblastoma

2. Contraindications of interventional tumor treatment should be added. 

Reviewer 2 Report

The manuscript is well written, concise enough. Clear description of the scientific background, study design and research objective of the review.

The results and conclusions are reported in a clear and exhaustive manner, underlining the major innovations of the submitted study.

Excellent performance of the work with care and dedication in the search.

Very accurate images.

Bibliographic references adapted and updated to the most recent scientific studies.
